# mRNA-LNP COVID-19 Vaccine Lipids Induce Complement Activation and Production of Proinflammatory Cytokines: Mechanisms, Effects of Complement Inhibitors, and Relevance to Adverse Reactions

**DOI:** 10.3390/ijms25073595

**Published:** 2024-03-22

**Authors:** Tamás Bakos, Tamás Mészáros, Gergely Tibor Kozma, Petra Berényi, Réka Facskó, Henriette Farkas, László Dézsi, Carlo Heirman, Stefaan de Koker, Raymond Schiffelers, Kathryn Anne Glatter, Tamás Radovits, Gábor Szénási, János Szebeni

**Affiliations:** 1Nanomedicine Research and Education Center, Department of Translational Medicine, Semmelweis University, 1085 Budapest, Hungary; tamas.bakos94@gmail.com (T.B.); tmeszaros@seroscience.com (T.M.); gkozma@seroscience.com (G.T.K.); berenyipetra@gmail.com (P.B.); facsko.reka.seroscience@gmail.com (R.F.); dr.dezsi.laszlo@gmail.com (L.D.); szenasi.gabor@med.semmelweis-univ.hu (G.S.); 2SeroScience LCC., 1089 Budapest, Hungary; 3Department of Cardiology, Heart and Vascular Center, Semmelweis University, 1122 Budapest, Hungary; radovitstamas@yahoo.com; 4Department of Surgical Research and Techniques, Heart and Vascular Center, Semmelweis University, 1089 Budapest, Hungary; 5Hungarian Center of Reference and Excellence, Department of Internal Medicine and Hematology, Semmelweis University, 1088 Budapest, Hungary; farkas.henriette@semmelweis.hu; 6Etherna Biopharmaceuticals, 2845 Niel, Belgium; carlo.heirman@etherna.be (C.H.); stefaan.dekoker@etherna.be (S.d.K.); 7Division of Laboratories and Pharmacy, University Medical Center, 3584 CX Utrecht, The Netherlands; schif101@gmail.com; 8Department of Education, Gratz College, Philadelphia, PA 19027, USA; kaglatter@gmail.com; 9Department of Nanobiotechnology and Regenerative Medicine, Faculty of Health Sciences, Miskolc University, 3530 Miskolc, Hungary; 10Translational Nanobioscience Research Center, Sungkyunkwan University, Suwon 06351, Republic of Korea

**Keywords:** anaphylatoxins, inflammation, serum, PBMC, complement inhibitors, Eculizumab, Berinert, endocarditis, pericarditis

## Abstract

A small fraction of people vaccinated with mRNA–lipid nanoparticle (mRNA-LNP)-based COVID-19 vaccines display acute or subacute inflammatory symptoms whose mechanism has not been clarified to date. To better understand the molecular mechanism of these adverse events (AEs), here, we analyzed in vitro the vaccine-induced induction and interrelations of the following two major inflammatory processes: complement (C) activation and release of proinflammatory cytokines. Incubation of Pfizer-BioNTech’s Comirnaty and Moderna’s Spikevax with 75% human serum led to significant increases in C5a, sC5b-9, and Bb but not C4d, indicating C activation mainly via the alternative pathway. Control PEGylated liposomes (Doxebo) also induced C activation, but, on a weight basis, it was ~5 times less effective than that of Comirnaty. Viral or synthetic naked mRNAs had no C-activating effects. In peripheral blood mononuclear cell (PBMC) cultures supplemented with 20% autologous serum, besides C activation, Comirnaty induced the secretion of proinflammatory cytokines in the following order: IL-1α < IFN-γ < IL-1β < TNF-α < IL-6 < IL-8. Heat-inactivation of C in serum prevented a rise in IL-1α, IL-1β, and TNF-α, suggesting C-dependence of these cytokines’ induction, although the C5 blocker Soliris and C1 inhibitor Berinert, which effectively inhibited C activation in both systems, did not suppress the release of any cytokines. These findings suggest that the inflammatory AEs of mRNA-LNP vaccines are due, at least in part, to stimulation of both arms of the innate immune system, whereupon C activation may be causally involved in the induction of some, but not all, inflammatory cytokines. Thus, the pharmacological attenuation of inflammatory AEs may not be achieved via monotherapy with the tested C inhibitors; efficacy may require combination therapy with different C inhibitors and/or other anti-inflammatory agents.

## 1. Introduction

The nucleoside-modified mRNA–lipid nanoparticle (mRNA-LNP)-based COVID-19 vaccines, BNT162b2 (Comirnaty) from Pfizer-BioNTech and mRNA-1273 (Spikevax) from Moderna, are the most widely applied vaccines against SARS-CoV-2 in the United States and many other countries worldwide. These vaccines are considered effective and safe [1,2]; nevertheless, like any medical intervention, a small fraction of vaccine recipients develop acute or subacute adverse events (AEs). Some of them are severe (SAEs), leading to emergency care, hospitalization, permanent disability, or even death. 

An international team of established experts (Brighton Collaboration’s Safety Platform for Emergency Vaccines) compiled a list of COVID-19 vaccine-specific AEs [3,4], whose incidence was analyzed in the Phase II/III placebo-controlled trials of mRNA-LNP vaccines. The more extensive Pfizer trial demonstrated a 36% higher risk for such “special interest” SAEs [3,4] in the vaccine group whose symptoms, importantly with respect to the present study, overlapped with those of acute and chronic COVID-19. 

These findings imply that the rare AEs caused by mRNA-LNP vaccines may have the same mechanism by which the SARS-CoV-2 virus causes harm, most likely a consequence of common structural features of mRNA LNPs and the virus. In other words, beyond its mission to code for S-protein (SP) expression by antigen-presenting immune cells, mRNA-LNP vaccines share with the virus the capability to cause certain pathologic effects in a small fraction of vaccinees.

Table 1 lists the AEs reported or claimed for COVID-19 mRNA vaccines, categorized by the organ systems affected. Importantly, myocarditis/pericarditis [5,6,7,8,9,10,11,12], HSRs/anaphylaxis [13,14,15,16], autoimmune diseases [17], thrombosis, thrombocytopenia and other coagulation disorders [18,19,20,21], skin [22,23,24,25] and ocular inflammations [26,27,28], Guillain–Barré syndrome [29,30,31,32,33,34,35], and other neurologic problems [36,37] can all be linked to acute or chronic inflammatory processes that are also characteristic of SARS-CoV-2 infections.

It is known that the pathogenesis of severe COVID-19 involves complement (C) activation, a phylogenetically ancient, basic defense mechanism [38,39,40,41,42,43,44], and also the release of proinflammatory cytokines, whose most intense manifestation is cytokine storm syndrome [45]. These processes represent the activation of the humoral and cellular arms of innate immunity, respectively, and are intricately interconnected in a vicious cycle [38,45]. Consequently, there are several studies attesting to the efficacy of C inhibitors against cytokine storm in severe COVID-19 [46,47,48,49,50]. Although less well known, C activation [14,16] and release of inflammatory cytokines [51,52,53] have also been described for mRNA vaccines and their LNP models; however, in the absence of awareness of the parallelism between the innate immune stimulatory effects of the virus and the vaccine, the recommendation for using C inhibitors to prevent “vaccine injury” has remained speculative to date [13]. Obtaining proof of principle requires a better understanding of the relationship between vaccine-induced C activation and the production of inflammatory cytokines; in particular, evidence for a causal relationship between the two processes is required. 

Accordingly, the main goals of the present study were to analyze the mechanisms of C activation and its role in cytokine release by Comirnaty and Spikevax and to explore the sensitivity of these processes to the clinically used C5 inhibitor Eculizumab and C1 inhibitor Berinert. We used human serum-supplemented PBMC cultures as a clinically relevant in vitro system for the simultaneous measurement of C activation and cytokine release [54,55] and 75% serum for the C-focused experiments. The latter included the analysis of activation pathways and the effects of C inhibitors and controls. Doxebo [56,57], i.e., placebo Doxil, was used as a PEGylated nanoparticle (NP) control because of the similarities in their nanostructures [56] and the wealth of information on C activation and related hypersensitivity/anaphylactoid reactions caused by Doxil [57]. Zymosan was used as the gold standard of C activation, thus enabling across-the-board conclusions.

## 2. Results

### 2.1. Complement Activation by Comirnaty in PBMC Culture

Figure 1A shows the levels of soluble terminal complex (sC5b-9) in PBMC supernatants as an endpoint of C activation after incubation with Comirnaty. In native serum-supplemented PBMC cultures (solid-colored bars), the averaged data from five different blood donors showed a significant rise in sC5b-9 at 45 min relative to the 0 min baseline both in the absence and the presence of Comirnaty, the rise being more pronounced in the presence of the vaccine. These data suggest that incubation at 37 °C for 45 min induces significant spontaneous C activation in PBMC, upon which the vaccine induced small activation superimposes. Surprisingly, we also recorded significant increases in sC5b-9 in heat-inactivated serum even at baseline, upon which the vaccine caused additional C activation relative to the vaccine-free 45 min samples. This counterintuitive observation is likely related to the liberation of heat-stable sC5b-9 upon heat treatment, which is superimposed on spontaneous C activation during incubation. This phenomenon obscured the difference between heated and native serum upon spontaneous C activation, but the Comirnaty-induced small, but significant, increase in sC5b-9 was still present. The latter data raise the possibility of incomplete decomplementation upon heat inactivation of serum, or cascade-independent (residual) C5 activation [58]. 

Figure 1B shows the Comirnaty-induced sC5b-9 changes in Panel A at the individual level. This presentation points to a biological, rather than a technical, cause of variation since one of the five sera (Subject #3) consistently displayed higher (up to 2-fold) sC5b-9 values compared with the other four sera under all test conditions. This donor may have had increased sensitivity towards C activation in general, exemplifying the common experience of substantial individual variation in immune response.

Taken together, these findings in 20% human serum confirm the previous data in 60% pig serum that Comirnaty can activate the C system [14]. Nevertheless, the dilution of serum reduced the sensitivity of the assay by 5-fold, thus reducing the effective dynamic range of sC5b-9 changes. This led us to test the cell-free native serum, which was diluted by 25% only with the added test agents. As shown below, this switch from 20 to 75% serum allowed us to validate the sC5b-9 assay for quantifying vaccine-induced C activation, to assess its pathways and relative potency, and to assess the effects of C inhibitors.

### 2.2. Features of C Activation by Comirnaty in 75% Human Serum

Figure 2A–C show the effects of the mRNA vaccine, Comirnaty (red bar), a PEGylated liposome, Doxebo (blue bar), and zymosan (black bar) on C activation in 75% sera of five blood donors (different donors than those shown in Figure 1), along with the time-matched PBS (baseline) values (yellow bars). To quantify C activation, we measured concurrent changes in the concentration of four reaction biomarkers including sC5b-9, a stable end product of the C cascade (Panel A); the anaphylatoxin C5a, one of the strongest proinflammatory mediators in blood (B); C4d, a C activation biomarker specific for classical and/or lectin pathway activations (C); and Bb, a biomarker specific for the alternative C pathway activation (D). 

It is seen in the figure that both Comirnaty and Doxebo caused small but significant 50–220% increases in sC5b-9 (Figure 2A), C5a (B), and Bb (D) compared with the PBS baseline. In contrast, C4d was significantly increased only by Doxebo, and the Comirnaty-induced smaller increase in C4d was seen only in three of the five sera. A closer analysis of these data also reveals that the Comirnaty-induced increase in sC5b-9 in the sera of different donors on the μg mL^−1^ scale (A) was proportional to the rise in C5a on the ng mL^−1^ scale (B). Furthermore, using the MWs specified in the legend, the mean concentration increases of ~4 μg/mL sC5b-9 and ~60 ng/mL C5a correspond to 4 and 6 nM, respectively, which ratio is a realistic deviation from equimolar formation of sC5b-9 and C5a, as only a part of the de novo formed C5b-9 binds to S-protein (vitronectin) in serum [59], which is measured by the sC5b-9 kit. The above facts, taken together with the ~4-fold higher Comirnaty-induced rise in sC5b-9 in 75% serum (Figure 2A) compared with that in 20% serum (Figure 1A), attest to the proportionality of C activation with the samples’ serum content in the two types of assay systems, validating the sC5b-9 assay for the quantification of C activation under different conditions. 

Further notable observations in Figure 2 are that Comirnaty and Doxebo caused far lower C activation, <1% of the activity of zymosan (Table 2), and that the Comirnaty- and Doxebo-induced rises in sC5b9, C5a, and Bb were near equal (Figure 2A,B,D), although, importantly, Comirnaty had ~6.2-fold less lipid in the samples than Doxebo (see legend). 

### 2.3. Pathways and Relative Efficacies of C Activation by Comirnaty and Doxebo

Beyond the quantification of C activation by Comirnaty and Doxebo, Figure 2 also provides information on the activation pathways of the two NPs. Notably, the finding that Doxebo caused a significantly higher increase in C4d than Comirnaty (Figure 2C) indicates that the classical pathway is more strongly involved in Doxebo- than Comirnaty-induced activation, and this difference was reduced in the Bb assay of the alternative pathway (D). The attenuation of this difference is consistent with alternative pathway amplification of classical C pathway activation, increasing the range of Bb changes to 4-fold higher concentration levels (see MWs in the legend). The proposal of primarily alternative pathway activation by Comirnaty was strongly supported by the highly significant correlation between the vaccine-induced increases in C5a and both sC5b-9 and Bb (Figure 3A,B), but not between C5a and C4d (Figure 3C).

Regarding the nearly identical increases in sC5b-9, C5a, and Bb caused by Comirnaty and Doxebo (Figure 2A,B,D), although the lipid amount was 6.2-fold lower in Comirnaty compared with Doxebo, this result raised the possibility of substantial differences in C activating efficacy. Quantifying the latter by dividing the vaccine-induced C activation, expressed as the maximum increase in sC5b-9 (*sTCC*), by the weight of putative C activating ingredient (i.e., lipid) (*CAI*), the resulting Specific C Activation (*SCA*) ratio was 4.6-fold higher in Comirnaty than in Doxebo (Table 2). 

### 2.4. C Activation by Spikevax 

To further dissect the individual contributions of vaccine ingredients to C activation, we also tested Spikevax, the other widely used COVID-19 mRNA vaccine, for C activation in the same 80% serum assay. The lipid composition of Comirnaty and Spikevax is different (see the legend in Figure 4); thus, if C activation was a property of a particular LNP lipid, it might be different for equivalent amounts of the two vaccines. However, as shown in Figure 4A, Comirnaty and Spikevax caused nearly identical low-level C activation in three human sera, a finding that was confirmed by another measurement performed as part of the inhibitor testing experiment (Figure 5B,C). These data suggest that independent of the core lipids, the PEGylated lipid membrane coating, a generic feature of mRNA-LNP vaccines, is one contributing factor to C activation.

### 2.5. Lack of C Activation by Naked mRNAs

Regarding the role of mRNA in the C activation by Comirnaty and Spikevax, we also measured and found negligible or no effect of SARS-CoV-2 mRNA in the experiment presented in Figure 4A (green bars). This result was confirmed in another experiment shown in Appendix A, suggesting that mRNA plays no role in the C activation by the two vaccines. However, it must be considered that the SARS-CoV-2 full mRNA contains 7-fold more nucleotides than the SP-encoding vaccine mRNA (29,903 vs. 4284 nucleotides) [56,60], and due to a lack of information on the structure–function relationship regarding nucleic acid-induced C activation, it cannot be a priori excluded that smaller naked mRNAs have different activating potentials. Therefore, we tested the C-activating effects of three other well-characterized, virus-independent mRNAs that were even smaller than the vaccines’ SP-mRNA. The experiment shown in Figure 4B indicates small, but statistically significant increases in sC5b-9 in two of the three mRNA samples tested, but only at a 100-fold higher mRNA level than present in the vaccine. Thus, these data strengthened the argument against an active role of mRNA in C activation by the vaccines. Nevertheless, there is a further theoretical limitation of the conclusion on the immune inertness of mRNA in the vaccine, as delineated in the discussion.

### 2.6. Effects of C Inhibitors on C Activation in PBMC Culture and 75% Sera 

Beyond the analysis of C activation by Comirnaty, a major goal of our study was to explore possibilities for the inhibition of this adverse immune reaction. To this end, we tested the efficacies of two clinically available C inhibitors including the C5 blocker Soliris and the C1 inhibitor Berinert. Figure 5A–E show two types of experiments iterating the test systems and different variables. In PBMC culture, Soliris, but not Berinert, exerted the complete inhibition of sC5b-9 formation in all five tested sera (Figure 5A). In the case of Soliris, this effect was reproduced in 75% serum for both Comirnaty- (Figure 5B) and Spikevax-induced (Figure 5C) C activation, while Berinert exerted partial inhibition of Comirnaty-induced C activation in two independent measurements of serum (Figure 5D,E). These data provide evidence for the efficacy of both C inhibitors against mRNA vaccine-induced C activation, Soliris being more effective than Berinert, at least at the tested concentrations.

### 2.7. Cytokine Production in Comirnaty-Exposed, Serum-Substituted PBMC: The Effects of Heat Inactivation and Inhibition of C Activation 

Figure 6A–F show significant induction by Comirnaty of the production of IL-1α, IL-1β, IL-6, IL-8, IFN-γ, and TNF-α, respectively, in the same heated or native 20% autologous serum-supplemented PBMC cultures that were tested for C activation in Figure 1B. In contrast, IL-2, IL-4, and IL-10 did not display any response to the vaccine. The secreted amounts of responder cytokines increased in the following order: IL-1α < IFN-γ < IL-1β < TNF-α < IL-6 < IL-8. Importantly, the increases in IL-1α, IL-1β, and TNF-α were absent in the heat-inactivated serum-supplemented samples, while those of IL-6 and IFN-γ were entirely serum-independent. The most abundantly produced IL-8, a chemokine that responds to most lipid- and polymer-based NPs [54], showed partial serum dependence, at least in four of the five individuals.

Since the heat inactivation of serum in the PCMB cultures did not reduce the production of IL-6 and IFN-γ (Figure 6C,E), cell starvation was unlikely to play a major role in the reductions in IL-1α, IL-1β, and TNF-α in the cultures supplemented with heated (C-depleted) serum. These data, therefore, suggest that the vaccine-induced production of at least the above three, but possibly some more yet untested, cytokines is C-dependent. To obtain alternative evidence for this conclusion and explore the utility of C inhibitors in inhibiting the C-dependent release of IL-1α, IL-1β, and TNF-α, we also tested the effects of Soliris and Berinert on cytokine induction at doses that inhibited C activation (Figure 5). However, unexpectedly, they did not reduce the production of any of the tested cytokines (Figure 6A–F). An interpretation of these data is in the Discussion below.

## 3. Discussion

Prompted by the rise in unusual (“special interest”) AEs of COVID-19 mRNA-LNP vaccines and consequent scientific and public debate over their safety, this study investigated the molecular mechanisms and relationship between two innate immune-stimulating effects of these vaccines that are assumed to jointly contribute to many independent SAEs of the vaccine (Table 1) including complement activation and production of proinflammatory cytokines. The in vitro data led to several new insights into the molecular mechanism of these phenomena, whose specifics and clinical relevance are discussed below.

### 3.1. Complement Activation by Comirnaty: Novel Findings and Mechanism 

Complement activation by liposomes and other therapeutic or diagnostic NPs has been known for decades, but the fact that mRNA-LNPs can also have such activity has not obtained much attention in the mainstream literature on COVID-19 vaccines. We have previously reported C activation caused by Comirnaty in pig serum in vitro [14] and in pig blood in vivo [16], and the present reproduction of the reaction in human serum implies that it is a species-independent phenomenon, fitting into the concept that C activation is an intrinsic property of PEGylated NPs due to their resemblance to pathogenic viruses [61]. 

Beyond some technical novelties, such as the stability of sC5b-9 during the heat treatment of serum and the lack of cellular damage in tissue culture supplemented with intact serum, the novel findings in the present study include (i) the dominance of alternative pathway activation, (ii) the increased strength of C activation relative to corresponding PEGylated liposomes, and (iii) the absence of C activation by naked mRNAs.

Regarding the occurrence of C activation by the vaccine, it was recently pointed out [13] that in theory, almost all components of Comirnaty have the capability to promote the activation cascade for different reasons. Thus, membrane-forming PEGylated lipids may activate C due to the binding of anti-PEG antibodies [16,62], DSPC, as a long, saturated fatty acid-containing unnatural phospholipid that rigidify, and cholesterol, which stabilizes the membrane coat for antibody and/or C3b binding. ALC-0315 may activate C because of its positive charge, enabling complexation with the mRNA63 and amphiphile character, leading to micelle and cluster formation [63], and, possibly, distribution into lipoproteins. 

The proposal of C activation by mRNA-ALC-0315 complexes is based on a previous study by Plank et al. [64] showing that DNA complexes with certain cationic lipids were strong C activators. The above study by Rissanou et al. also presented evidence for cluster formation from positively charged lipids63 which, in analogy to positively charged liposomes [65], could be another source of C activation via the alternative pathway. It is not excluded, either, that the liberated ALC-0315 associates with plasma lipoproteins entailing C3-binding and alternative pathway C activation, in a manner like that observed with Cremophor EL [66]. 

Regarding the ~5-fold higher “specific” result, i.e., weight-normalized C reactivity of Comirnaty vs. Doxebo (Table 2), the finding implies that the vaccine has a C-activating feature that Doxebo does not have. The crucial difference is the presence of mRNA and positively charged lipids in the vaccine, so the multiple possibilities by which the mRNA and core lipids could activate C could account for the increased C reactivity of the vaccine despite the lack of C activation by naked mRNAs. However, to contribute to C activation, the internal content of the vaccine needs to be released into the blood. Meet this precondition can also be predicted based on the mentioned recent structural analysis of Comirnaty [56], claiming that the vaccine consists of soft, fragile NPs prone to break up in water [56]. In fact, the latter study showed electron microscopic images of lipid fragments and snake-like twisted nanostructures in disintegrated Comirnaty samples, which were tentatively identified as mRNA-lipid complexes [56]. It should also be mentioned that the vaccine-induced significant increase in Bb (Figure 2D) and its correlation with the rises in C5a and sC5b-9 (Figure 3) indicate the predominance of alternative pathway C activation, which can most easily be rationalized by exposure of the Comirnaty’s core components to serum, i.e., rapid disintegration.

### 3.2. Inflammatory Cytokine Production by Comirnaty: Mechanism and Relationship with C Activation

The other Comirnaty-induced proinflammatory process analyzed in the present study as a possible contributing mechanism to the vaccine-induced SAEs was the induction of inflammatory cytokines and its dependence on C activation. A crosstalk between these two arms of the innate immune system has already been described in many previous studies [55,67,68,69,70], but not for mRNA-LNP vaccine-induced immune stimulation. In theory, Comirnaty can enhance cytokine release both via direct interaction with immune cells and their indirect stimulation via anaphylatoxin and other C receptors, and our PBMC data (Figure 6) gave an indication of the operation of both mechanisms. The clinically relevant, novel finding in our study was the identification of intact serum-dependence of the production of IL-1β, IL-6, TNF-α, and IF-γ, which provides indirect evidence for the C-dependence of the release of these cytokines. The resultant proposal that C activation plays a causal role in this process is supported by an earlier PBMC study showing the suppression of zymosan-induced IL-6 release by the alternative pathway C inhibitor, mini-Factor H [55]. 

### 3.3. Broader Implications of In Vitro Data on C Activation and Cytokine Induction by mRNA-LNP Vaccines

#### 3.3.1. The Complement Test as an In Vitro Model for Vaccine-Induced Anaphylaxis

The detection of C activation by the vaccine supports the concept that anaphylaxis, listed among the most frequent SAEs of Comirnaty [71], represents C activation-related pseudoallergy (CARPA) [13]. Although Comirnaty is injected i.m. at a much lower lipid dose than applied upon CARPAgenic i.v. liposome therapies [61], there are considerations that reconcile the differences in administration. First, the vaccine can rapidly exit from the deltoid muscle into the blood via several routes [13]; thus, similarly to i.v. liposomes, it may activate C in blood. Second, the weight-based in vitro C reactivity of Comirnaty was ~5 times stronger than that of Doxebo, whose reactivity is close to that of Doxil in causing CARPA [57]. Consequently, the threshold concentration to cause CARPA in reactive individuals may be significantly lower for Comirnaty. Third, the incidence of mRNA-LNP vaccine-induced anaphylaxis (<0.04%) [15] is comparable or lower than that estimated for Doxil (~1% of <11% ≤ 0.1%) [72], a realistic proportionality between anaphylaxis incidence and estimated C activation in vivo.

#### 3.3.2. Use of the PBMC-Based Pan-Innate Immune Stimulation Assay for Finding Effective Pharmacological Prevention of mRNA-Vaccine-Induced SAEs

We found that serum inactivation, assumably because of C depletion, does not necessarily prevent cytokine release; for some cytokines, it was effective, and for others, it was not. We therefore tested the effects of Soliris and Berinert, two clinically used C inhibitors, as an alternative approach to modify C activity. Although they inhibited C activation, they did not inhibit the production of IL-1α, IL-1β, or TNF-α, thus contradicting the causal relationship between C activation and secretion of these “serum-dependent” cytokines. Although we did not analyze further the possible reasons for these negative results, one explanation may be the mentioned fact that Comirnaty may enhance cytokine release by the simultaneous activation of multiple surface receptors, in which case, the inhibition of classical pathway activation by Berinert, or C5a production by Soliris, may remain sub-threshold for cytokine induction. Thus, the single use of either of these inhibitors may not be enough to attenuate the inflammatory reaction; effective prevention may require simultaneous inhibition of the C cascade at multiple checkpoints, for which there are numerous C inhibitor drugs and drug candidates available [73,74,75,76,77]. The double-endpoint PBMC assay applied in the present study therefore offers a simple in vitro model for testing the efficacy of C inhibitors against inflammatory SAEs. 

#### 3.3.3. Public Health Implications

By showing the occurrence of two essential proinflammatory effects by mRNA-LNP vaccines, the present study attempts to give a scientific explanation for the “special interest” SAEs that these vaccines rarely cause, yet that induce substantial press and media attention with top-level public debates in the U.S. Senate and U.K. Parliament in the context of vaccine safety and excess death. The number of publications on different COVID-19 vaccine-induced SAEs has been on the rise [5,6,7,8,9,10,11,12,13,14,15,16,17,18,19,20,21,22,23,24,25,26,27,28,29,30,31,32,33,34,35,36,37], and once the real scale and mechanism of these AEs are better understood, the focus can be redirected to the prevention and, hence, further extension of mRNA-LNP technology for the development of new vaccines and gene therapies [78,79,80]. The present study contributes to the latter goal.

## 4. Materials and Methods

### 4.1. Materials

Mammalian cell culture medium (RPMI-1640, R5), non-essential amino acid solution (0.1 mM), pyruvate solution (1 mM), penicillin–streptomycin antibiotics solution and β-mercaptoethanol (50 µM) Ficoll-Paque, ethylenediaminetetraacetic acid (EDTA), Dulbecco’s phosphate buffered saline (D-PBS), and trypan blue dye were purchased from Merck Life Science Ltd. (Budapest, Hungary). R5 medium was from ThermoFisher Scientific. Comirnaty and Spikevax vaccines used against the delta variant of the SARS-CoV-2 virus (lot numbers: ET7205 and FA 5829) were obtained from Semmelweis University’s Pharmacy. They were stored as instructed by the manufacturers and immediately used after opening the vials within the expiration date. The C inhibitor anti-C5 monoclonal antibody (mAb), Eculizumab (Soliris^®^), was from Alexion Pharmaceuticals (Boston, MA, USA), today owned by Astra Zeneca (Cambridge, U.K.). The C1-esterase inhibitor (Berinert^®^) was from CSL Behring GmbH (Marburg, Germany), and 5-methoxyuridine-modified SARS-CoV-2 spike protein mRNA (E484K, N501Y) was from OZ Biosciences SAS (Marseille, France). 

### 4.2. Ethical Permission, Donors, and Blood Collection

The Scientific and Research Ethics Committee of the Medical Research Council of Hungary granted ethical approval for this research project (TUKEB 15576/2018/EKU). After obtaining informed consent from all white Caucasian, COVID-19 vaccinated, healthy adult volunteers of this study (total n = 23) their blood was withdrawn by a phlebotomist.

### 4.3. Separation of Serum and PBMC

The serum was separated from whole blood after coagulating the blood (approx. 25 min) at room temperature in a Greiner VACUETTE Z Serum Sep Clot Activator (8 mL) blood collector tube. The tubes were centrifuged (2000× *g*, 4 °C, 15 min), and the supernatants (serum) were collected. PBMCs were separated from uncoagulated whole blood collected in Greiner VACUETTE K2E K_2_EDTA (9 mL) blood collector tubes and diluted in a 1:1 ratio with 6 mM EDTA containing D-PBS using Ficoll-Paque density gradient centrifugation (500× *g*, 25 °C, 30 min). Cells that formed a white-colored ring were then suspended in 6 mM EDTA containing D-PBS to ensure the removal of the remaining substances from the previous steps and were centrifuged again (500× *g*, 25 °C, 30 min). To remove thrombocytes and residual EDTA, cells were washed with cold, EDTA-free, D-PBS and were centrifuged again (500× *g*, 4 °C, 15 min). Following the centrifugation, the cells were divided into three portions and were suspended in 3 culture media as follows: (1) medium only (R5); (2) R5 containing 20% autologous human serum, representing the in vivo conditions with 1/5 the amount of C proteins and other serum components exposed to the cells; (3) R5 containing 20% autologous human serum inactivated by heat, representing in vivo conditions without activable C proteins. To inactivate sera, also known as decomplementation, a portion of the innate serum from each donor was incubated at 56 °C for 30 min. The 20% serum ratio was chosen based on preliminary studies showing no significant cell toxicity over 18 h, as determined by trypan blue exclusion. The suspensions were centrifuged (500× *g*, 4 °C, 15 min) and subsequently re-suspended in the culture media specified above. 

### 4.4. Preparation of mRNAs 

To investigate the C-activating effects of naked mRNAs, we applied N1-Methyl Pseudouridine-modified, 5′ cap1 cleancapped mRNAs prepared by Etherna Biopharmaceuticals (Niel, Belgium), as described earlier [81]. The samples coding for the proteins CD40L (29.3 kDa), Cre-recombinase (38.5 kDa), and luciferase (60.7 kDa) contained 1200, 1450, and 2086 nt, respectively. Their sequence and other details are shown in the Appendix A. 

### 4.5. PBMC Studies

Following separation, the PBMCs were suspended in R5 medium supplemented with 20% native or heat-inactivated autologous serum. The suspension was then placed into the inner wells of 96-well microtiter plates, with each well containing approximately 5 × 10^5^ cells. The plates were placed in a CO_2_ incubator (5% CO_2_) at 37 °C. Aliquots were collected at 45 min and 18 h after the start of the experiment. Additionally, baseline samples (“0 min samples”) were collected before the incubation. After collection, the samples were centrifuged (2500× *g*, 4 °C, 10 min), and the supernatants were stored at –80 °C until the C and cytokine assays were performed. Soluble terminal C complex (*sTCC*, sC5b-9) was measured with an ELISA kit from Svar Life Science AB (Malmö, Sweden), and for the measurement of cytokines, we used a Q-View™ LS chemiluminescent imager paired with Q-View™ Software, https://www.quansysbio.com/products-and-services/imaging/q-view-imager-ls/ (accessed on 18 March 2023), obtained from Quansys Biosciences (West Logan, UT, USA). The cytokine panel supplied by the company was the Human Cytokine Inflammation Panel 1 multiplex ELISA, measuring IL-1α, IL-1β, IL-2, IL-4, IL-6, IL-8, IL-10, IFN-γ, and TNF-α. All procedures followed the manufacturer’s instructions. 

### 4.6. Serum Studies 

The vaccine and other test samples were mixed with native serum at a 1:3 volume ratio, followed by incubation at 37 °C for 30 or 45 min, as stated in the text. The incubation was stopped by dilution with the kit’s sample diluent supplemented with 10 mM EDTA. Soluble TCC, C5a, C4d, and Bb were measured with Svar’s C5b-9 kit (Svar Life Sci., Malmö, Sveden) and TECO*medical*’s C5a, C4d, and Bb ELISA kits (TECO*medical* AG, Sissach, Switzerland).

### 4.7. Statistical Analysis 

All data reported are means ± SEM. Statistical comparisons were made using paired *t*-tests or one-way ANOVA and Dunnetts’ or Tukey’s multiple comparisons post hoc tests, as specified in the text.

## Figures and Tables

**Figure 1 ijms-25-03595-f001:**
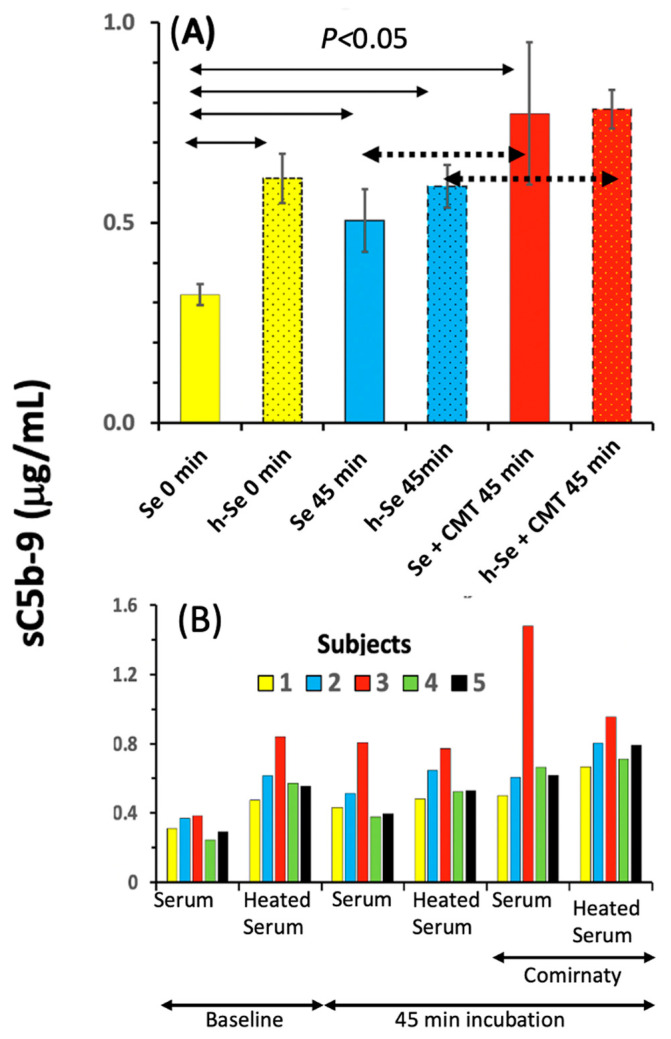
Complement activation by Comirnaty in PBMC cultures in R5 medium supplemented with 20% native (solid-colored bars) or heat-inactivated autologous serum (dash-framed dotted bars). The final concentration of the vaccine was 16 μg mRNA/mL. Bars in (**A**) show averages (±SEM) from 5 donors and yellow, blue, and red colors indicate baseline and 45 min incubation in the absence and presence of Comirnaty, respectively. Statistical comparisons were performed using repeated measures ANOVA followed by Dunnett’s multiple comparisons post hoc tests. Panel (**B**) shows the same data at the individual level. The incubation in a CO_2_ incubator at 37 °C lasted for 18 h, while aliquots were taken for the sC5b-9 ELISA after 45 min. The arrows indicate the significance of differences, dotted arrows pair the bars that show vaccine-induced C activation either in native or heat-inactivated serum-supplemented cultures. Se, serum; h-Se, heated serum, 0 min: before incubation, after adding the serum; 45 min, after 45 min incubation; CMT, Comirnaty.

**Figure 2 ijms-25-03595-f002:**
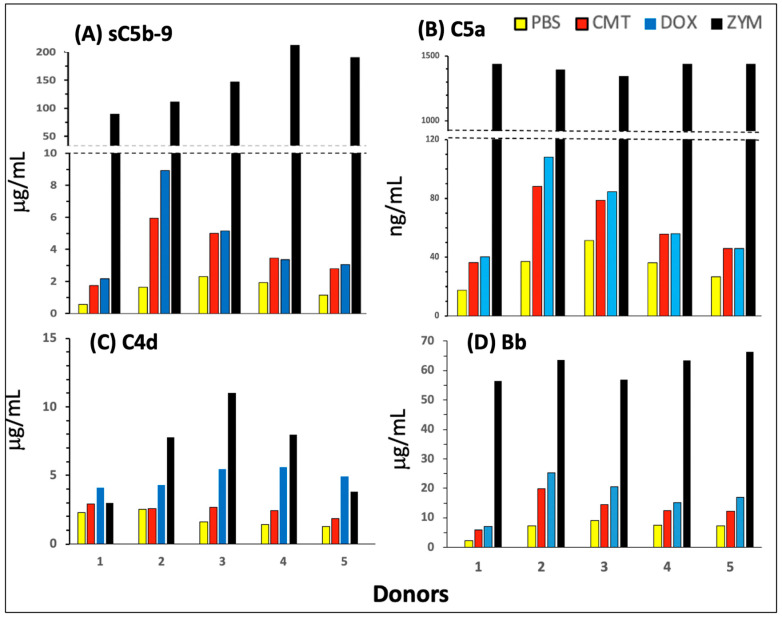
Complement activation by Comirnaty (CMT), control PEGylated liposomes (Doxebo, DOX), and zymosan (ZYM) in 75% serum of 5 blood donors who were different from those in Figure 1. The total lipid in the Comirnaty and Doxebo samples were 0.64 and 3.99 mg/mL, respectively (Appendix A), ZYM was applied at 0.3 mg/mL, and the mRNA concentration in Comirnaty was 25 μg mRNA/mL. The above lipid contents were based on equal, 4-fold dilution of vaccine and liposome stocks in serum and imply 6.2-fold higher lipid concentrations in Doxebo than that in Comirnaty. Bars with different colors represent different reaction triggers, as defined in the key. The individual rises relative to baseline were significant for all 4 reaction markers (**A**–**D**) by paired 2 sample *t*-tests at *p* < 0.05 or <0.01. Conversion of the average Comirnaty-induced increases in C4d (~1 μg/mL, (**C**)) and Bb (~10 μg/mL, (**D**)) to molar concentrations (the MWs of C4d and Bb were 47 and 60 kDa, respectively) gave a C4d/Bb ratio of 1:7.

**Figure 3 ijms-25-03595-f003:**
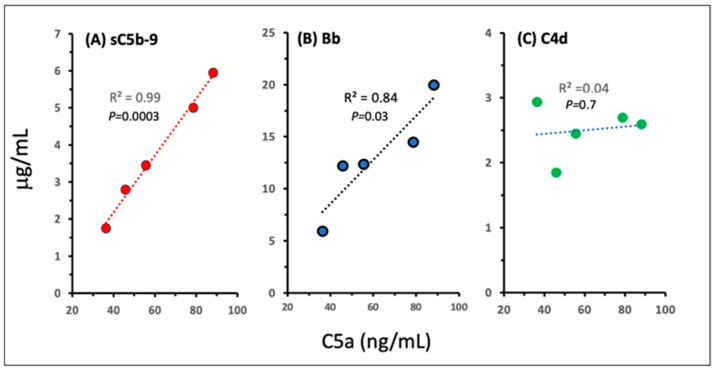
Correlations between C5a release and other reaction markers during Comirnaty-induced C activation in 75% human serum. These data were derived from the experiment also shown in Figure 2. The R^2^ and *p*-values near the regression lines indicate the significance of the correlation.

**Figure 4 ijms-25-03595-f004:**
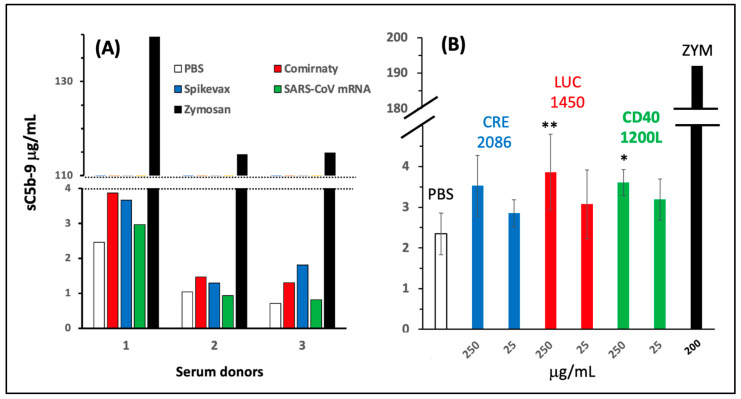
Complement activation by Comirnaty, Spikevax, and SARS-CoV-2 mRNA (**A**) and synthetic, virus-irrelevant mRNAs (**B**) in human serum. The experiments were similar to those in Figure 2A, except that the blood donors were different. In A, the final mRNA concentrations in the Comirnaty, Spikevax, and SARS-CoV-2 samples were 20 μg/mL. The lipids in Comirnaty are (((4-hydroxybutyl) azanediyl) bis (hexane-6,1-diyl)bis(2-hexyldecanoate))(ALC-0315), (2-((polyethylene glycol)-2000)-N,N-ditetradecylacetamide) (ALC-0159), 1,2-distearoyl-sn-glycero-3-phosphocholine (DSPC), and cholesterol (total lipids: 2.57 mg mL^−1^). The relative amounts of these Comirnaty ingredients are specified in Appendix A. The lipids in Spikevax are heptadecan-9-yl 8-((2-hydroxyethyl) (6-oxo-6-(undecyloxy)hexyl)amino) octanoate (SM-102), polyethylene glycol-2000-dimyristoyl glycerol (PEG-DMG), DSPC, and cholesterol (total lipids: 3.86 mg mL^−1^). In B, the mRNAs tested were codings for the proteins CD40L, Luciferase, and Cre-recombinase, transcribed from the moCD40L, rstsFluc, and Cre-recombinase genes. Their sequences are shown in Appendix A, and the nucleotide numbers are on top of the bars. The abscissa shows the concentrations of these mRNAs added to serum: ZYM, zymosan. Mean ± SEM, n = 5 different sera. *, **, significant differences compared with PBS at *p* < 0.05 and *p* < 0.01 (ANOVA followed by Dunnetts’ post hoc test).

**Figure 5 ijms-25-03595-f005:**
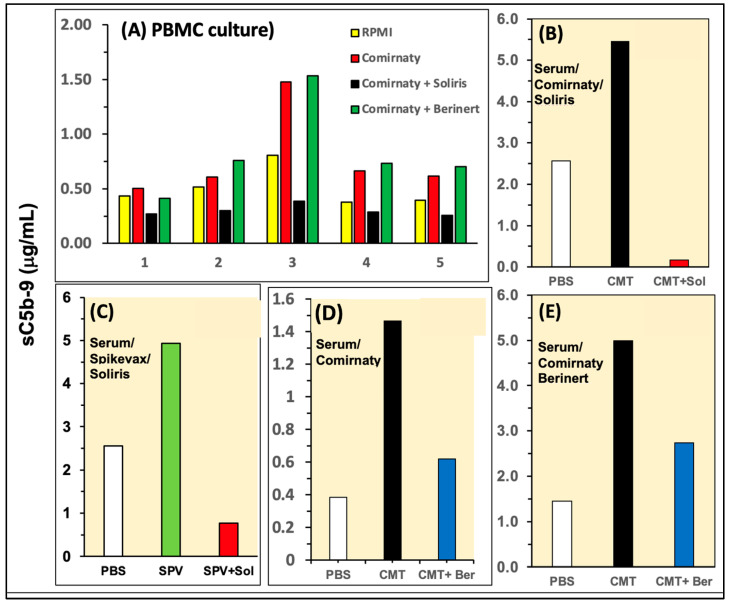
(**A**) Inhibition of Comirnaty-induced C activation by Soliris in PBMC cultures. The experiment is similar to that in Figure 1B. (**B**–**E**) Experiments in 75% serum using the C activators and inhibitors specified beside the bars. Yellow bars in (**A**) and empty bars in (**B**–**E**) show the serum baselines.

**Figure 6 ijms-25-03595-f006:**
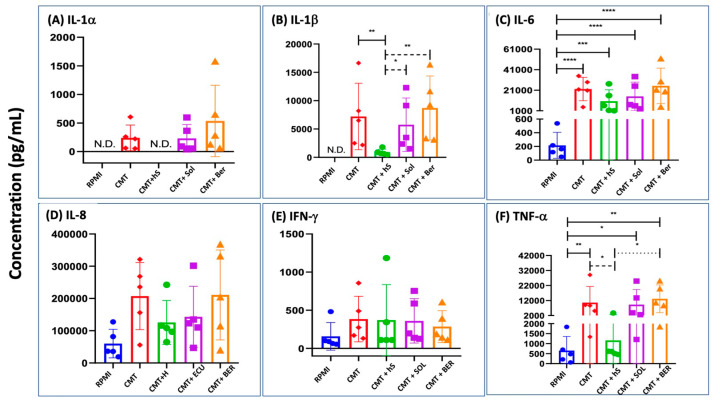
Cytokine production by Comirnaty in PBMC culture supplemented with 20% autologous serum with or without heat inactivation and C inhibitors. Differently colored symbols from 5 blood donors distinguish the treatments. The concentrations of Comirnaty, Soliris (SOL), and Berinert (BER) were 16 μg/mL (mRNA), 1 mM, and 1 mM, respectively. Bars show the mean along with the individual points. Statistical comparisons were performed using one-way ANOVA followed by Tukey’s post hoc tests. *, **, ***, and **** mean *p* < 0.5, 0.1, 0.05, and 0.01, respectively. Other experimental conditions are described in the Methods. N.D., non-detectable.

**Table 1 ijms-25-03595-t001:** Recognized and claimed adverse effects of mRNA-LNP vaccines *.

**Symptoms**	**Cardiovascular**	**Coagulation**	**Enteral**	**Immune**	**Neural**	**Respiratory**	**Skin**
	ST elevation/AMI	cerebral venous sinus thrombosis	hepatitis	anaphylaxis	encephalitis/ myelitis/ encephalomyelitis	ARDS	acute urticaria
	tachy- or bradycardia, arrhythmias	disseminated intravascular coagulation	cholecystitis	hypersensitivity reactions	Bell’s palsy	pulmonary embolism	chronic urticaria
	vascular inflammation (Kawasaki disease)	immune thrombocytopenia	colitis	lymphadenopathy (Kawasaki disease)	Guillain–Barré syndrome	stridor, hoarseness	skin graphia, dermatographia
	myocarditis/ pericarditis	pulmonary embolisms	enteritis	autoimmune glomerulonephritis	narcolepsy/ catalepsy	dyspnea	dermatographic urticaria
	hypo/ hypertension	stroke (hemorrhagic/ ischemic)	diarrhea	autoimmune rheumatic disease	seizures/ convulsions/ epilepsy	coughing	rash
	stroke (hemorrhagic/ ischemic)	thrombosis with thrombocytopenia (VITT)	appendicitis	autoimmune hepatitis	transverse myelitis		ocular/orbital inflammation
	arteriosclerosis	venous thrombo-embolism		CARPA	delirium		
	chest/back pain	amenorrhea/ dysmenorrhea/ oligomenorrhea			akathisa (psychomotor restlessness)		
	other forms of cardiac injury	thrombocytopenic purpura			multiple sclerosis		
	lip, tongue, face edema	intracerebral hemorrhage					
	death	fibrous white clots ^†^					

* Compilation of AEs addressed in the literature; refs. [5,6,7,8,9,10,11,12,13,14,15,16,17,18,19,20,21,22,23,24,25,26,27,28,29,30,31,32,33,34,35,36,37] or claimed in public media and still under debate. Abbreviation: ST, ST segment in the ECG; AMI, acute myocardial infarction; WITT, vaccine-induced immune thrombocytopenia and thrombosis; CARPA, complement activation-related pseudoallergy; ^†^ Thrombus, formed by the aggregation of platelets, fibrin and, possibly, amyloid peptides. A debated AE.

**Table 2 ijms-25-03595-t002:** Comparison of C activation by Comirnaty and Doxebo in 75% human serum.

Test Agent	sTCC (sC5b-9, μg/mL) *		CAI	SCA (TCC/CAI) ***	Zym % ^#^	CMT/Dox ^##^
	Mean	SEM (n = 5)	(mg/mL) **			
**Comirnaty**	**2.3**	**0.6**	0.5	4.6	0.9	4.6
Doxebo	3.1	1.1	3.2	1.0	0.2	
Zymosan	148.4	22.9	0.30	494.7	100.0	

The experiments were similar to those shown in Figure 2 and Figure 3, except that the 5 serum donors were different. * *sTCC*, mean ± S.E.M. of baseline-corrected sC5b-9 readings after incubation with the test agents (Column 1) for 30 min; **, *CAI*, final concentration of C-Activating Ingredient (mg lipid/mL), which is the sum of different lipid concentrations in Comirnaty and Doxebo vials (vial stock concentrations are in Appendix A), and the dry weight/mL, in the case of zymosan; ***, *SCA*, Specific C Activation (mean *sTCC/CAI*); ^#^
*Zym%*, *SCA* related to Zym; ^##^ CMT/Dox *SCA* ratio, where CMT is Comirnaty and Dox is Doxebo.

## Data Availability

Data will be made available upon request.

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
