# Peer review of "mRNA-LNP COVID-19 Vaccine Lipids Induce Complement Activation and Production of Proinflammatory Cytokines: Mechanisms, Effects of Complement Inhibitors, and Relevance to Adverse Reactions"

_ijms, 2024, doi:10.3390/ijms25073595_

Round 1

Reviewer 1 Report

Comments and Suggestions for Authors

The manuscript entitled "mRNA-LNP COVID-19 vaccine lipids induce complement activation and production of proinflammatory cytokines: Mechanisms, effects of complement inhibitors, and relevance to adverse reactions" submitted by Bakos et.al discusses the identification of the common factor present in the COVID19 vaccine induced adverse effects and the COVID19 virus induced inflammatory effects. This identification of the common factor will shed light on the mechanism by which we see the adverse effects in the vaccines which will eventually help in the designing of better vaccine strategies.  In order to study this the authors focuses on the two major mediators of inflammation i.e. complement activation and the production of pro-inflammatory cytokines. 

The authors have a clear hypothesis and to test this the author used appropriate experimental design. The manuscript is well written, and the statistical methods used are appropriate to the best of my knowledge. The material and methods sections provide complete information. The discussion is well written and shows the future perspective. However, there are some concerns as follows

The authors should keep the same format for the cytokines on the line 36 and 37 eg. IL-1β

For the Figure 1 A the authors should mention about the method by which they obtained the p values in the legends section as they did elsewhere in the other Figure legends. 

The work done by Bakos et.al is commendable.

Reviewer 2 Report

Comments and Suggestions for Authors

ijms-2888276

mRNA-LNP COVID-19 vaccine lipids induce complement ac-tivation and production of proinflammatory cytokines: Mechanisms, effects of complement inhibitors, and relevance to adverse reactions

The manuscript by Bakos et al. reported that the mRNA-LNP vaccine, Comirnaty, triggers low level complement (C) activation and production of inflammatory cytokines. The study was appropriately conducted and the manuscript was suitable for publication after revision. Please consider the following comments.

1. The authors should clarify the research gap and highlight the novelty and contribution of this study.

2. The study rationale (section 3.2) should be moved to the Introduction.

3. Figure 3/ left panel: Please rearrange to avoid hiding the X-axis.

4. Figure 5 legend: Please use A, B, C, D, and E instead of Figure 5A-E.

5. Please provide the demographic summary of donors.

6. The number of references (130) is too high for a research article and needs to be lowered. Please remove unnecessary references if possible.

Comments on the Quality of English Language

Minor editing of English language required

Reviewer 3 Report

Comments and Suggestions for Authors

This manuscript presents intriguing content; however, several issues detract from its clarity and scientific rigor. Firstly, the abstract lacks a clearly stated aim and conclusion. The introduction concludes with a mention of "our findings," whereas it should articulate the hypothesis or aim. There appears to be a lack of coherence between the Introduction and the Results sections.

The manuscript is generally disorganized, necessitating thorough editing to enhance readability and comprehension.

Specifically e.g., Table 1 has a small font size; not explained abbreviations used throughout the document, including but not limited to PBMC and COVID-19.

Regarding English usage, revision is required to facilitate understanding. For instance, the phrase “Cytokine production by Comirnaty in PBMC culture supplemented with 20% autologous serum” misleadingly implies that cytokines are produced by the Comirnaty vaccine, rather than by PBMCs from patients vaccinated with Comirnaty. This exemplifies the broader issue of clarity of the manuscript, highlighting the need for detailed clarification of technical descriptions.

In the "materials and methods" should clearly state how many patients the biological material came from. Have they had a previous COVID vaccination? Were the same sera used in all experiments?

Round 2

Reviewer 2 Report

Comments and Suggestions for Authors

The manuscript was appropriately revised and can be accepted.

Reviewer 3 Report

Comments and Suggestions for Authors

The authors have satisfactorily responded to the comments. Thank you.